# Genetic requirement for *Esrp1* and *Esrp2* in vertebrate pituitary morphogenesis

Shannon H. Carroll[1,2], Sogand Schafer[1], Ariella S. Richman[1], Peng Wang[3], Mian Umair Ahsan[3], Lisa Tsay[1], Kai Wang[3,4] and Eric C. Liao[1,2,*]

## ABSTRACT

The pituitary gland produces several hormones that regulate growth, metabolism, stress response, reproduction and homeostasis. Congenital hypopituitarism is a deficiency in one or more pituitary hormones and encompasses a spectrum of clinical conditions. The pituitary has a complex embryonic origin, with the oral ectoderm contributing the anterior lobe, and the neural ectoderm generating the posterior lobe. Pituitary abnormalities and growth deficiencies are associated with cleft palate; however, the developmental genetic connection between pituitary and orofacial cleft malformations remains to be determined. The epithelial RNA splicing regulators *Esrp1* and *Esrp2* (*Esrp1/2*) are required for orofacial development in zebrafish, mice and humans, and loss of function of these genes results in a cleft palate. Here, we present a detailed developmental analysis of the genetic requirement for *Esrp1/2* in pituitary morphogenesis in mouse and zebrafish. Further, we describe an individual with cleft palate and hypopituitarism who harbors a nucleotide variant in the RNA-binding domain of *ESRP2*. The discovery of this key function for *Esrp1/2* in pituitary formation has significant fundamental and clinical implications for understanding congenital hypopituitarism and craniofacial anomalies.

**KEY WORDS: *ESRP1*, *ESRP2*, Cleft palate, Pituitary, Adenohypophysis**

## INTRODUCTION

The pituitary gland is a midline structure that resides in the sella turcica recess of the sphenoid bone at the base of the brain. The pituitary has a complex embryonic origin, with the oral ectoderm contributing the anterior lobe, the neural ectoderm generating the posterior lobe and a recently identified contribution from the endoderm (Fabian et al., 2020; McCabe and Dattani, 2014). The pituitary gland produces several hormones with essential systemic physiological effects in maintaining growth, metabolism, stress response, reproduction and homeostasis (McCabe and Dattani, 2014; Prodam et al., 2021; Rizzoti and Lovell-Badge, 2005).

[1]Center for Craniofacial Innovation, Division of Plastic and Reconstructive Surgery, Department of Surgery, Children's Hospital of Philadelphia, Philadelphia, PA 19104, USA. [2]Shriners Hospital for Children, Tampa, FL 33607, USA. [3]Center for Cellular and Molecular Therapeutics, Children's Hospital of Philadelphia, Philadelphia, PA 19104, USA. [4]Department of Pathology and Laboratory Medicine, University of Pennsylvania, Philadelphia, PA 19104, USA.

*Author for correspondence (liaoce@chop.edu)

 E.C.L., 0000-0001-6385-7448

Congenital hypopituitarism (CH) is defined as a deficiency in one or more pituitary hormones and encompasses a spectrum of clinical conditions with both familial and sporadic occurrences (Higham et al., 2016; McCabe and Dattani, 2014; Prodam et al., 2021). Non-specific symptoms may present in neonates. However, the full clinical sequalae of central pituitary deficiencies may not present until adolescence or young adulthood (Higham et al., 2016).

Midline craniofacial defects, including holoprosencephaly (HPE) and septo-optic dysplasia (SOD), are often associated with pituitary deficiency (McCabe and Dattani, 2014; Prodam et al., 2021). Pituitary abnormalities and growth deficiencies are also associated with isolated cleft palate (Akin et al., 2014; Bowers et al., 1988; Laron et al., 1969; Roitman and Laron, 1978; Rudman et al., 1978), a common birth defect occurring in ∼1 in 700 births (Dixon et al., 2011). The developmental genetic connection between pituitary and orofacial cleft malformations remains to be determined.

We and others have shown that the epithelial RNA splicing regulators *Esrp1* and *Esrp2* are required for orofacial development in zebrafish, mice and humans (Bebee et al., 2015; Burguera et al., 2017; Carroll et al., 2020). Ablation of *esrp1* and *esrp2* in zebrafish results in several developmental defects, including a cleft that involves the upper mouth opening and the anterior neurocranium (ANC) (Burguera et al., 2017; Carroll et al., 2020), a structure with embryonic similarity to the mammalian primary palate (Carroll et al., 2020). In the mouse, ablation of *Esrp1* causes a bilateral cleft lip and cleft palate, and ablation of both *Esrp1* and *Esrp2* leads to a more severe bilateral cleft lip and palate phenotype (Bebee et al., 2015; Carroll et al., 2020; Lee et al., 2020). Finally, *ESRP2* human gene variants have been found in cleft lip and palate clinical cohorts, whereas, so far, *ESRP1* gene variants have been associated with hearing deficits (Cox et al., 2018; Rohacek et al., 2017).

Across zebrafish and mouse, we found that *Esrp1* and *Esrp2* transcripts (*Esrp1/2*) are colocalized in epithelial cells throughout early embryonic development, including in the surface and oral epithelium (Carroll et al., 2020). Since the oropharyngeal epithelium contributes to anterior pituitary development, we hypothesized that pituitary development may be disrupted in *Esrp1/2* null murine and zebrafish models. Herein, we present a detailed developmental analysis of *Esrp1/2* in mouse and zebrafish revealing the genetic requirement for *Esrp1/2* in pituitary morphogenesis. Given the key role of *Esrp1/2 in* generating specific isoforms of its target genes that regulate epithelial function, the discovery of this key role for *Esrp1/2* in pituitary formation has significant fundamental and clinical implications for understanding CH and craniofacial anomalies.

## RESULTS

### *Esrp1* and *Esrp2* transcripts are detected in Rathke's pouch, and in the anterior and intermediate pituitary

We described previously that *Esrp1* and *Esrp2* genes are expressed within the oral epithelium of mouse embryos (Carroll et al., 2020).

Previous work has shown that the mammalian anterior and intermediate pituitary develop through an invagination of the oral epithelium, termed Rathke's pouch. We analyzed *Esrp1* and *Esrp2* expression during key time points of mouse pituitary morphogenesis. At E10.25, *Esrp1* and *Esrp2* transcripts were colocalized in the surface epithelial cells lining the cranial structures and the oral epithelium, including the invaginating epithelium that will go on to form Rathke's pouch (Fig. 1A). No *Esrp1/2* gene expression was detected within the neuroectoderm. At E11.5 and E12.5, *Esrp1* and *Esrp2* were colocalized in the enclosed epithelium, forming Rathke's pouch (Fig. 1B and C, respectively). *Esrp1* and *Esrp2* gene expressions persisted in the mature pituitary gland (P0) (Fig. 1D), particularly in the marginal zone and to a lesser degree in the anterior pituitary parenchyma. The expression of *Esrp1/2* across the span of pituitary morphogenesis suggested a potential role in morphogenesis.

### Anterior pituitary failed to develop in *Esrp1/2* null mice

Previous work showed that *Esrp1* null and *Esrp1/2* compound null mice exhibit bilateral cleft lip and cleft palate, that and neonates die shortly after birth (Bebee et al., 2015; Carroll et al., 2020; Lee et al., 2020). To investigate the role of *Esrp1/2* in pituitary morphogenesis, we collected pups from *Esrp1^{+/−}; Esrp2^{+/−}* breeding in-crosses. Histological examination of the pituitary gland and surrounding structures at P0 and E17.5 revealed that *Esrp1^{−/−}; Esrp2^{−/−}* compound null mice exhibited severely hypoplastic or absent anterior pituitary (Fig. 2 and Fig. S1A, respectively). We observed a similar phenotype in *Esrp1^{−/−}; Esrp2^{+/+}* pups at E17.5, where the anterior pituitary was absent (Fig. S1B). As *Esrp1^{+/+}; Esrp2^{−/−}* mice grow and reproduce normally, we did not analyze pituitary development in the *Esrp2* nulls. The intermediate pituitary of *Esrp1^{−/−}; Esrp2^{−/−}* null mice is dysmorphic, whereas the posterior pituitary is intact, though somewhat hypoplastic (Fig. 2A). We confirmed that *Esrp1^{−/−}; Esrp2^{−/−}* null mice have a present, though dysmorphic, intermediate pituitary by staining for Sox2 expression, which is highly enriched in intermediate pituitary cells (Cheung et al., 2017) (Fig. 2B).

Growth hormone (GH) is expressed in the anterior pituitary and was absent in the *Esrp1^{−/−}; Esrp2^{−/−}* null mice (Fig. 2B). POMC is expressed in the intermediate and anterior pituitary, and was observed to be present but diminished in the *Esrp1^{−/−}; Esrp2^{−/−}* null mice (Fig. 2B). Taken together, these results indicate an absence of an anterior pituitary in *Esrp1^{−/−}; Esrp2^{−/−}* null mice, as well as the absence of functional anterior pituitary endocrine cells.

To determine the stage at which anterior pituitary development becomes disrupted in the *Esrp1^{−/−}; Esrp2^{−/−}* null mice, histology was performed at various time points. At E10.5, control embryos have significant invagination of the oral epithelium with a thickened layer of epithelium lining what will become Rathke's pouch. In *Esrp1^{−/−}; Esrp2^{−/−}* null littermates, there is reduced invagination and the epithelium remains a thin single layer (Fig. 3A). Consistent with reduced invagination of the oral epithelium, at E11.5, Rathke's pouch of *Esrp1^{−/−}; Esrp2^{−/−}* null embryos is smaller and dysmorphic, where closure of Rathke's pouch is incomplete as compared to littermate controls (Fig. 3B). At E12.5, control embryos have a fully formed Rathke's pouch that has separated from the oral epithelium (Fig. 3C, Fig. S2). In *Esrp1^{−/−}; Esrp2^{−/−}* null littermates, Rathke's pouch has invaginated and closed; however, the closure of the pouch is abnormal, and the resulting structure is dysmorphic and hypoplastic (Fig. 3C, Fig. S2).

During Rathke's pouch morphogenesis, a transcriptional program is activated that leads to the specification and differentiation of anterior pituitary progenitors and, ultimately, hormone-expressing cells (Zhu et al., 2007). To determine whether this program is disrupted in *Esrp1^{−/−}; Esrp2^{−/−}* null mice, *in situ* hybridization was performed at several developmental timepoints (Fig. 3A-C). The presence of *Isl1*, *Pitx2*, *Sox2*, *Lhx3* and *Prop1* mRNA in Rathke's pouch of *Esrp1^{−/−}; Esrp2^{−/−}* null mice suggested that the specification of pituitary progenitor cells was preserved, but subsequent proliferation and contribution of these cells to form the Rathke's pouch and anterior pituitary were defective (de Moraes et al., 2012; Gregory and Dattani, 2020; Perez Millan et al., 2024) (Fig. 3B, Fig. S2).

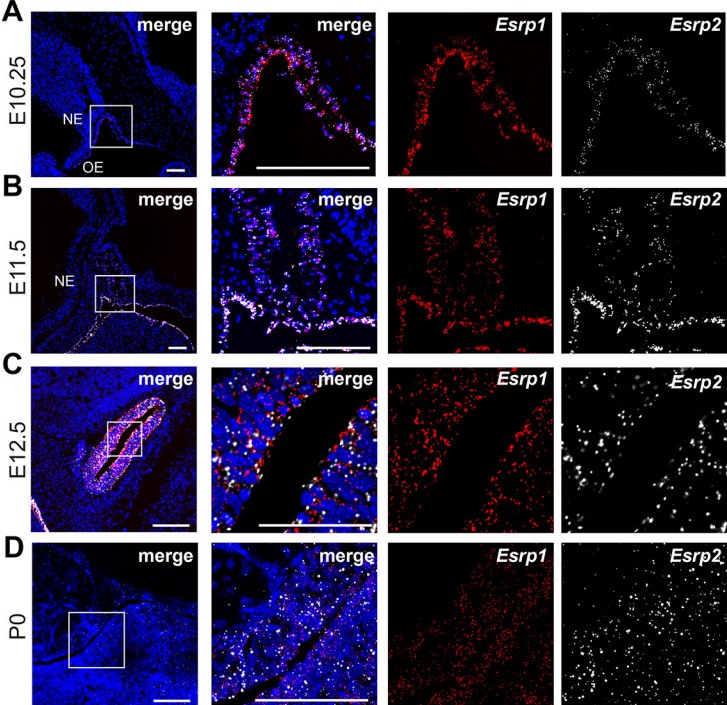

**Fig. 1. *Esrp1* and *Esrp2* are expressed in the developing mouse anterior pituitary.** (A-C) Sagittal sections of wild-type mouse Rathke's pouch at E10.25 (A), E11.5 (B) and E12.5 (C) analyzed by RNAscope *in situ* hybridization. Embryos are oriented with rostral to the left and dorsal at the top. Images on the left are at low magnification, while images on the right are higher magnifications of the outlined areas. Staining shows cellular co-expression of *Esrp1* (red) and *Esrp2* (white) in the invaginating oral epithelium and in the developing Rathke's pouch (A,B). (C) *Esrp1* and *Esrp2* expression persist after the closure of Rathke's pouch. (D) *Esrp1* and *Esrp2* continue to be expressed in the anterior and intermediate pituitary gland of mice at birth (P0). Nuclei are stained with DAPI (blue). *n*=3. Scale bars: 100 μm.

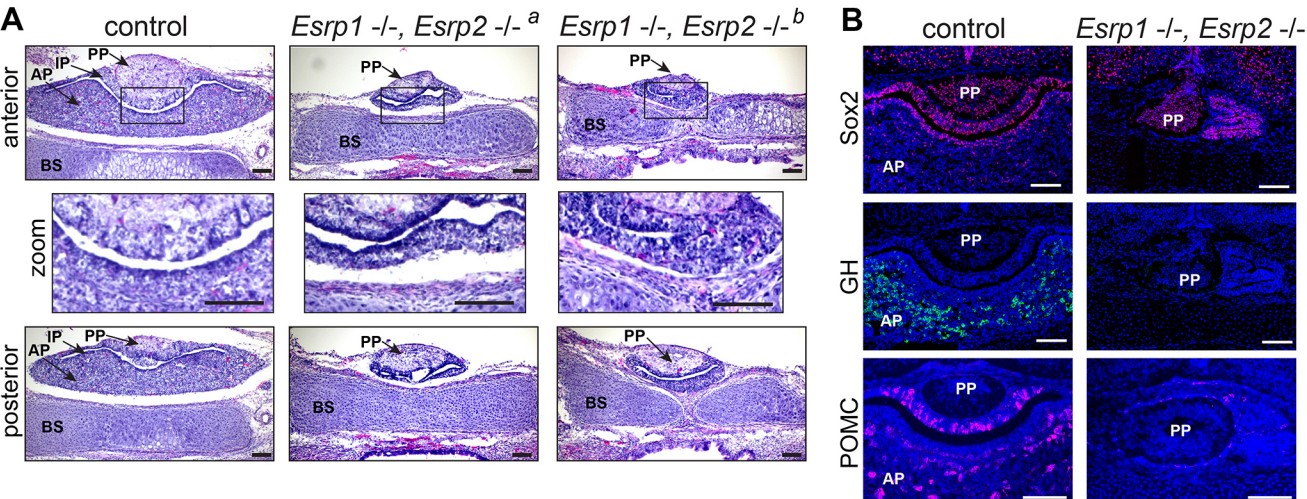

**Fig. 2. *Esrp1*−/−,*Esrp2*−/− null mice have an absent or severely hypoplastic anterior pituitary.** (A) Coronal Hematoxylin and Eosin stained sections of P0 littermate control (*Esrp1* +/+, *Esrp2* +/−) and *Esrp1*−/−, *Esrp2*−/− null pituitary. The top row shows rostral-positioned sections, the middle row shows an enlargement of the outlined areas and the bottom row shows caudal-positioned sections. *Esrp1*−/−*Esrp2*−/− null mice have a severely hypoplastic or absent anterior pituitary. The intermediate pituitary is somewhat intact, while the posterior pituitary appears unaffected. [a] and [b] indicate different individuals. (B) Coronal sections of E17.5 littermate control (*Esrp1* +/+, *Esrp2* +/−) and *Esrp1*−/−,*Esrp2*−/− null pituitary immunostained for Sox2 (magenta), growth hormone (GH, green) or POMC (magenta). Remnant pituitary tissue of *Esrp1*−/−,*Esrp2*−/− null mice highly expresses the pituitary stem cell marker Sox2, similar to the expression in the control intermediate pituitary; however, no GH expression and minimal POMC expression was detected. PP, posterior pituitary; IP, intermediate pituitary; AP, anterior pituitary; BS, basisphenoid. *n*=3. Scale bars: 100 μm.

Invagination of the oral epithelium and formation of Rathke's pouch relies on a series of signaling pathways and morphogens that are spatially restricted (de Moraes et al., 2012; Rizzoti and Lovell-Badge, 2005; Takuma et al., 1998; Treier et al., 1998; Zhu et al., 2007). One important signal for the invagination of the epithelium is Fgf secreted from the adjacent neuroectoderm (Ericson et al., 1998; Norlin et al., 2000; Treier et al., 1998). *Esrp1/2* are known to regulate the alternative splicing of Fgf receptors and are required for the expression of the epithelial Fgfr1IIIb isoform (Bebee et al., 2015). To test whether loss of *Esrp1/2* affects *Fgfr1* isoform expression in the invaginating epithelium forming Rathke's pouch, we performed Basescope isoform-specific *in situ* hybridization in E10.5 *Esrp1*−/−;*Esrp2*−/− null and littermate control embryos. As shown in Fig. 3, E10.5, *Esrp1*−/−; *Esrp2*−/− null mice have a smaller area of invaginated epithelium that is shallower than in control littermates (Fig. 4A). Further, *Esrp1*−/−; *Esrp2*−/− null mice have fewer cells exhibiting a columnar and stratified morphology, as compared to the littermate controls. The presence of Fgfr1IIIb and Fgfr1IIIc isoform transcripts were quantified by counting the number of individual Basescope signals detected within the invaginated epithelium. *Esrp1*−/−; *Esrp2*−/− null mice had lower total expression of the Fgfr1IIIb isoform as well as a lower number of isoform transcripts per cell (Fig. 4B). These data show that *Esrp1/2* regulates the *Fgfr1* epithelial isoform expression within the epithelium contributing Rathke's pouch. However, how changes to the *Fgfr1* isoform expression affect Rathke's pouch formation remains to be determined.

Although Rathke's pouch of *Esrp1/2*−/− embryos is hypoplastic, we would expect these cells to contribute some anterior pituitary cells. However, *Esrp1/2*−/− have no apparent anterior pituitary cells, leading us to hypothesize an additional defective mechanism. The further development of Rathke's pouch into the anterior pituitary lobe requires the rostro-ventral expansion of the progenitors lining the pouch. This expansion is thought to depend on an epithelial-to-mesenchymal transition (EMT) of the epithelial cells lining the pouch, which ultimately contribute the mesenchymal-like cells of

the developing gland (Perez Millan et al., 2016; Winningham and Camper, 2023; Zhu et al., 2007). Esrp1 and Esrp2 are known modulators of EMT (Göttgens et al., 2016; Warzecha et al., 2010), and their loss disrupts EMT in normal development (Warzecha et al., 2010). We found higher Cdh1 (E-cadherin) mRNA expression and protein expression in the ventral aspect of Rathke's pouch in *Esrp1/2*−/− embryos (Fig. 5A,B, respectively). As downregulation of Cdh1 is thought to be required for the movement of cells out of Rathke's pouch and into the anterior pituitary lobe (Himes and Raetzman, 2009), we conjecture that dysregulated EMT contributes to the hypoplastic or absent anterior pituitary of *Esrp1/2*−/− mice.

## *esrp1* and *esrp2* genes are expressed in the zebrafish adenohypophyseal placode and developing pituitary

It has been previously described that zebrafish compound mutants for *esrp1* and *esrp2* have abnormal craniofacial development (Burguera et al., 2017; Carroll et al., 2020). The upper margin of the mouth of *esrp1/2* mutant zebrafish is discontinuous and the anterior neurocranium exhibits a cleft in the median region, which is populated by an aberrant cluster of cells (Burguera et al., 2017; Carroll et al., 2020). These findings in zebrafish mutants corroborate that the developmental processes of orofacial cleft pathogenesis, with regard to the anterior structures such as the mouth opening and primary palate, are similar between zebrafish and amniotes.

Given the commonalities of the craniofacial phenotypes in *Esrp1/2* null mice and zebrafish, as well as the dysmorphic Rathke's pouch and hypoplastic anterior pituitary lobe observed in the *Esrp1*−/−,*Esrp2*−/− null mice, we examined pituitary morphogenesis in the *esrp1/2* mutant zebrafish. We first characterized *esrp1* expression in the developing adenohypophysis (ADH) of zebrafish embryos. At 24 h post-fertilization (hpf), the adenohypophyseal placode has specified at the rostral-most midline of the embryo and is identified by *pitx3* expression (Dutta et al., 2005; Zilinski et al., 2005). *esrp1* was highly expressed within these *pitx3*-expressing cells (Fig. 6A). At 27 hpf,

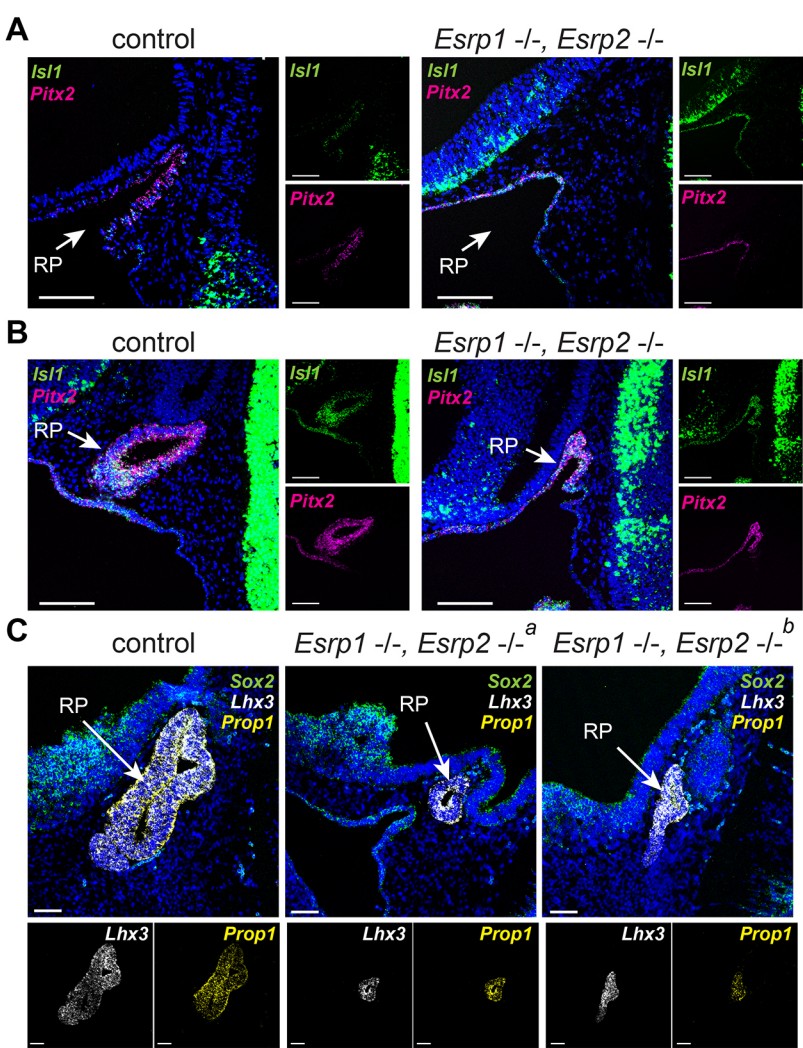

**Fig. 3. *Esrp1*<sup>−/−</sup>,*Esrp2*<sup>−/−</sup> null mice have similar expression of early pituitary differentiation factors, despite Rathke's pouch being hypoplastic and dysmorphic.** (A-C) Sagittal sections of Rathke's pouch in littermate control (*Esrp1* <sup>+/+</sup>, *Esrp2* <sup>+/−</sup>) and *Esrp1*<sup>−/−</sup>, *Esrp2*<sup>−/−</sup> null mouse embryos. (A,B) RNAscope *in situ* hybridization showing *Isl1* (green) and *Pitx2* (magenta) expression in E10.5 (A) and E11.5 (B) embryos. (C) RNAscope *in situ* hybridization showing *Sox2* (green), *Lhx3* (white) and *Prop1* (yellow) expression. <sup>a</sup> and <sup>b</sup> denote different individuals. Nuclei are stained with DAPI (blue). RP, Rathke's pouch. *n*=3. Scale bars: 100 μm.

the cells of the adenohypophyseal placode have begun moving caudally (Fig. 6B). At this early stage of development, presumptive adenohypophyseal and hypothalamic cells express pro-opiomelanocortin a (*pomca*) (Fig. 6B). *esrp1* transcripts were detected within *pomca*-expressing adenohypophyseal cells and cells of the developing oral epithelium (Fig. 6B). Therefore, similar to mouse, *esrp1/2* are expressed in the developing pituitary across morphogenesis.

### *esrp1/2* are required for zebrafish adenohypopyseal morphologenesis

To examine whether adenohypophysis development is disrupted in *esrp1/2* mutant zebrafish, whole-mount RNA *in situ* hybridization was performed for key genes that function in the pituitary, *pomca*, prolactin (*prl*) and growth hormone (*gh*; *gh1*), in *esrp1/2* mutant and clutch-mate controls at 3 days post-fertilization (dpf). Interestingly, in the *esrp1/2* mutants, cells marked with *pomca*, *prl* and *gh* transcripts were displaced ventro-rostrally relative to controls. *pomca* gene expression was particularly aberrant in *esrp1/2* mutants relative to controls, as *pomca*-positive cells were found to be lining the oral cavity, including the lower jaw element (Fig. 7A). Using RNAscope whole-mount RNA *in situ* hybridization, we obtained similar results for *pomca* expression. *esrp1/2* mutants have similar hypothalamic expression of *pomca* relative to controls, but there is also a high level

of expression within the mouth (Fig. 7B). Further, the expression pattern of *pomca* in the *esrp1/2* mutant mouths also varied more between individuals, in contrast to wild type, where the localization of *pomca* expression was consistent at the skull base (Fig. 7B).

To obtain cellular resolution of pituitary cells in control versus *esrp1/2* mutant zebrafish, we performed RNAscope *in situ* hybridization on sagittal sections obtained at the midline at 3 dpf. We used keratin 4 (*krt4*) expression to identify embryonic epithelium, and *lhx3* and *pitx3* gene expressions to define the adenohypophysis. Consistent with whole-mount imaging, we found that *lhx3* and *pitx3* transcripts were present in *esrp1/2* mutants but ectopically localized to a ventro-rostral position, suggesting a lack of transit to a more dorsal and caudal location where the pituitary would reside (Fig. 7C). *prl* and *pomca* were detected in the ectopically localized cells (Fig. 7C), which is analogous to the observation in the mouse that the pituitary progenitors are specified but fail to localize and contribute the anterior pituitary. Taken together, we show that the adenohypophysis of *esrp1/2* mutant zebrafish forms aberrantly, with pituitary progenitors failing to reach their proper location, though differentiation is unaffected.

*In situ* hybridization results suggest impaired translocation of the pituitary anlage in *esrp1/2* mutants. To visualize pituitary development in real-time, we generated a zebrafish pituitary reporter line by inserting tdTomato into the first exon of *lhx3*

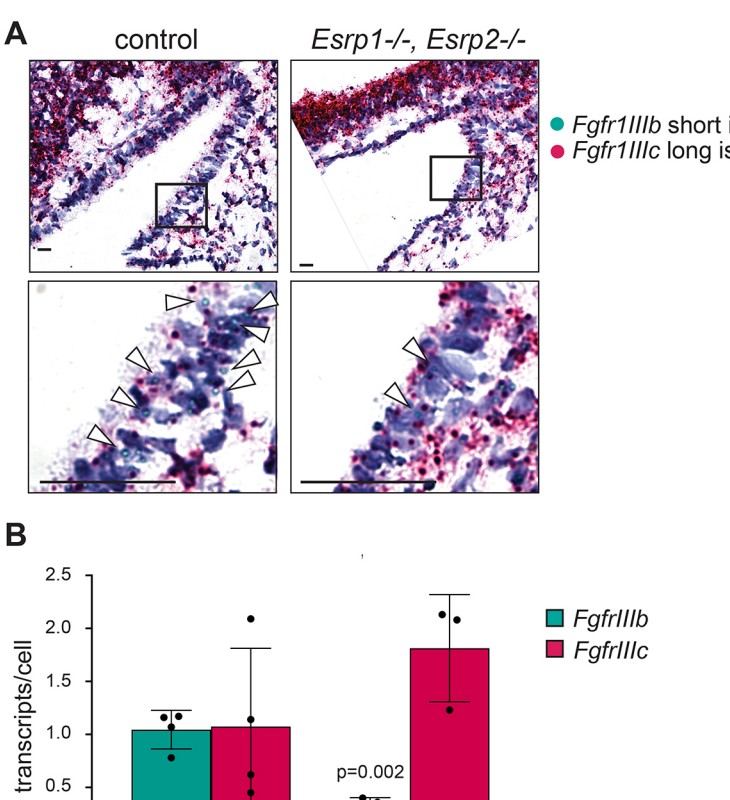

**A**

control *Esrp1-/-, Esrp2-/-*

● *Fgfr1IIIb* short isoform
● *Fgfr1IIIc* long isoform

**B**

**Fig. 4. *Esrp1*<sup>−/−</sup>,*Esrp2*<sup>−/−</sup> null mice express less Fgfr1IIIb isoform within the invaginating Rathke's pouch.**
(A,B) Sagittal sections and Basescope *in situ* hybridization of Rathke's pouch in E10.5 littermate control (*Esrp1*<sup>+/+</sup>, *Esrp2*<sup>+/+</sup>) and *Esrp1*<sup>−/−</sup>, *Esrp2*<sup>−/−</sup> null mouse embryos. (A) Representative images showing Rathke's pouch, where *in situ* hybridization was performed for Fgfr1IIIb (green) and Fgfr1IIIc (red) isoforms. Magnification of outlined area where staining for Fgfr1IIIb transcripts is indicated by arrowheads. Scale bars: 10 μm. (B) Quantification of the number of *Fgfr1* isoform transcripts normalized to the total number of nuclei within Rathke's pouch. *P*=0.002 (unpaired Student's *t*-test), *n*=3 biological replicates. Data are mean±s.d.

(Fig. 8A). In zebrafish, *lhx3* is one of the first genes expressed in the adenohypophyseal placode and pituitary anlage, and is highly tissue specific at the developmental time points of interest (Glasgow et al., 1997; Herzog et al., 2003). Light-sheet microscopic imaging of live zebrafish wild-type embryos from 27-42 hpf shows translocation of *lhx3*-positive cells from a position immediately superficial to the surface ectoderm near the future stomodeum to a ventral and more caudal position (Fig. 8B,C, Movies 1-4). The *lhx3*-expressing cells of the pituitary anlage also express *cdh1*, as was shown previously (Fabian et al., 2020).

We utilized our newly generated *lhx3*:tdTomato reporter line to visualize the developing pituitary in the *esrp1/2* mutant zebrafish. As *esrp1/2* mutants are visually identifiable starting at 48 hpf, we imaged mutants and wild-type clutch-mate controls at 48 and 72 hpf. Similar to our *in situ* hybridization results, we found that the *lhx3*-positive pituitary of *esrp1/2* mutants failed to translocate to the proper position and that *lhx3* expression was expressed exogenously in the oral cavity immediately superficial to the abnormal ANC (Fig. 8D,E).

### A single nucleotide variant in *ESRP2* is associated with cleft palate and hypopituitarism

Our finding of anterior pituitary dysmorphogenesis in *Esrp1/2* null mice and zebrafish, which also display cleft of the lip and palate, supports an etiological link between orofacial clefts and congenital hypopituitarism that has been previously proposed (Akin et al., 2014; Bowers et al., 1988; Laron et al., 1969; McCabe and Dattani, 2014; Prodam et al., 2021; Roitman and Laron, 1978; Rudman et al., 1978). To assay whether *ESRP2*, which has been previously reported to be associated with orofacial clefts (Cox et al., 2018; Rohacek et al., 2017), is also associated with congenital

hypopituitarism, we analyzed clinical and genomic data in the Arcus database from the Children's Hospital of Philadelphia. The Arcus database contains whole-exome sequencing data from 6903 individuals undergoing clinical exome sequencing testing for a variety of clinical conditions. Of these, we identified eight individuals with both congenital hypopituitarism and orofacial cleft. In one individual, we identified a nonsynonymous single nucleotide variant (SNV) (16-68232468-C-T in GRCh38 coordinate) in *ESRP2* that results in an arginine-to-histidine amino acid change (p.R286H) in the coding region of exon 8 based on MANE select transcript NM_024939.3 (Fig. 9A,B). This amino acid is conserved across species (Fig. 9C). This SNV is located in the RNA recognition motif domain, and it is predicted to be deleterious by the dbNSFP database (e.g. it has an AlphaMissense score of 0.95 and a MetaRNN score of 0.85) (Liu et al., 2020). It has a population maximum frequency of 0.0000147 in the gnomAD version 4.1 database (Karczewski et al., 2020), indicating that it is extremely rare. To examine whether there is another stronger candidate in the individual that explains the cleft palate, we examined the 418 previously known genes associated with orofacial cleft (Caetano da Silva et al., 2024), plus the *ESRP1* gene in this individual. However, we did not find another variant that is predicted to be deleterious or reported to be pathogenic in the ClinVar database (Landrum et al., 2014). Moreover, this *ESRP2* variant was predicted to be the most deleterious SNV based on dbNSFP database among all variants in the 419 genes. The known pathogenic variants reported in ClinVar for this individual were also examined. There are three known pathogenic variants in *PERM1* (ClinVar Variation ID: 1320032), *ITPKB* (ClinVar Variation ID: 1705896) and *SORD* (ClinVar Variation ID: 929258). None of these three variants can explain the

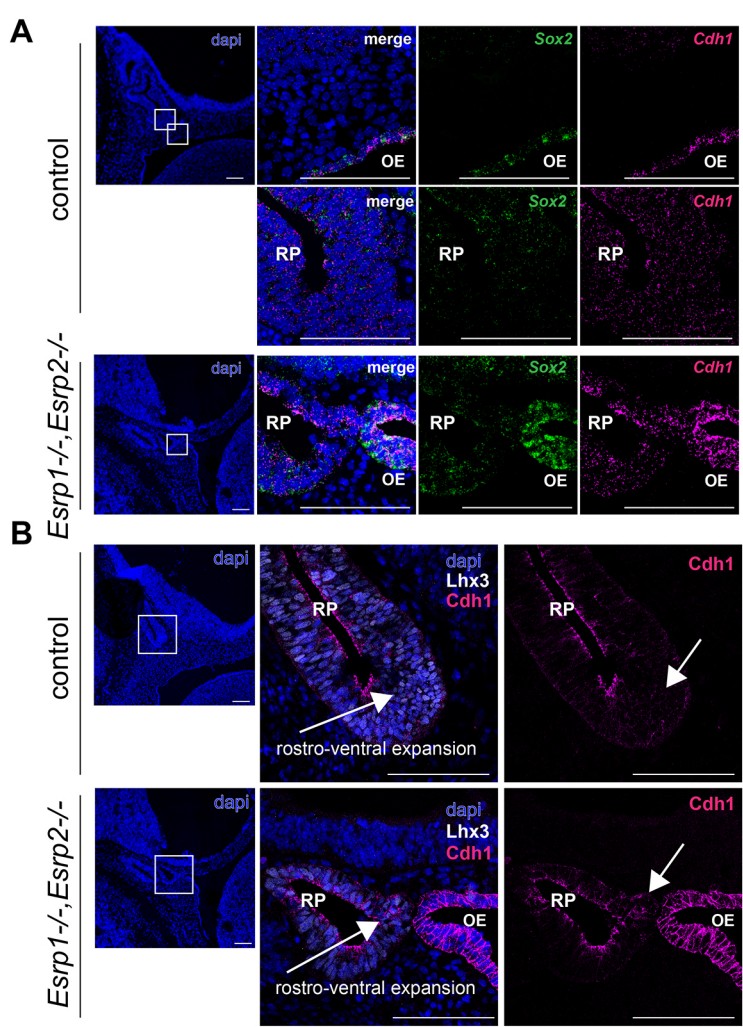

**Fig. 5. Cdh1 expression is higher in Rathke's pouch of *Esrp1*⁻/⁻, *Esrp2*⁻/⁻ null mice.** (A,B) Sagittal sections of *Esrp1*⁻/⁻, *Esrp2*⁻/⁻ and wild-type littermate control E12.5 embryos showing the onset of rostro-ventral expansion and migration of pituitary progenitors from Rathke's pouch (RP). Low magnification is shown on the left; higher magnifications of the outlined areas are shown on the right. (A) RNAscope *in situ* hybridization shows increased expression of Sox2 and Cdh1 in RP and the oral epithelium (OE) in *Esrp1*⁻/⁻, *Esrp2*⁻/⁻ embryos. (B) Immunofluorescence staining shows *Esrp1*⁻/⁻, *Esrp2*⁻/⁻ embryos have higher expression of Cdh1 (arrows), suggesting impaired epithelial-to-mesenchymal transition. Scale bars: 100 μm.

phenotype of pituitary deficiency and orofacial clefts. Therefore, the *ESRP2* variant remains the most likely pathogenic variant in this individual that explains both pituitary abnormality and orofacial cleft.

## DISCUSSION

Development of the mammalian pituitary gland requires an interaction between closely associated ectodermal tissue, namely the surface/oral ectoderm, and the neuroectoderm. Molecular signaling between these tissues must be spatially and temporally restricted, and coordinated such that specific alterations in ectoderm morphology are produced, resulting in the surface/oral ectoderm-derived anterior lobe and the neuroectoderm-derived posterior lobe. The events leading to pituitary organogenesis in zebrafish are less clear; however, several morphogens and growth factors have been shown to be conserved (Herzog et al., 2003, 2004; Pogoda and Hammerschmidt, 2007).

Here, we demonstrate that *Esrp1/2* are required for the early events of pituitary organogenesis in both mice and zebrafish. Early during surface/oral ectoderm invagination, *Esrp1*⁻/⁻;*Esrp2*⁻/⁻ mouse embryos demonstrate a more shallow invagination and abnormal cellular morphology of Rathke's pouch. The resulting

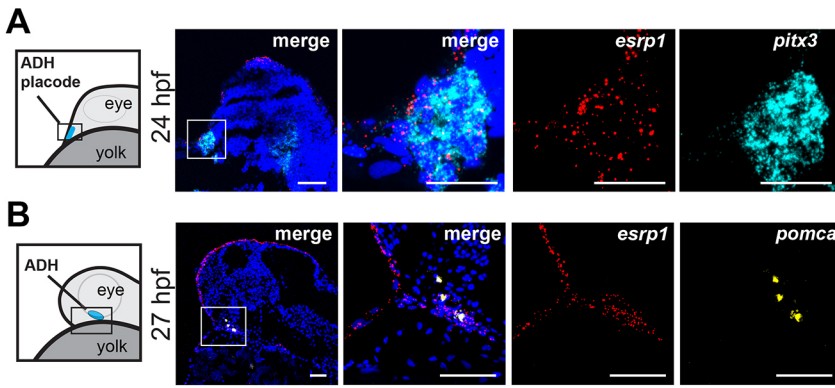

**Fig. 6. Zebrafish adenohypophysis placode and developing adenohypophysis express *esrp1*.** (A,B) Midline sagittal sections of wild-type zebrafish embryo at 24 hpf (A) and 27 hpf (B), analyzed by RNAscope *in situ* hybridization. Embryos are oriented with rostral to the left and dorsal at the top. Images on the right are increased magnifications of the outlined areas on the left. (A) At 24 hpf, *esrp1* (red) is expressed in the adenohypophysis (ADH) placode, which was identified by *pitx3* (cyan) expression. (B) At the time of posterior migration (27 hpf), *esrp1* is expressed in the developing ADH, identified by *pomca* expression (yellow). *esrp1* expression is also expressed in the developing oral epithelium. *n*=2. Scale bars: 50 μm.

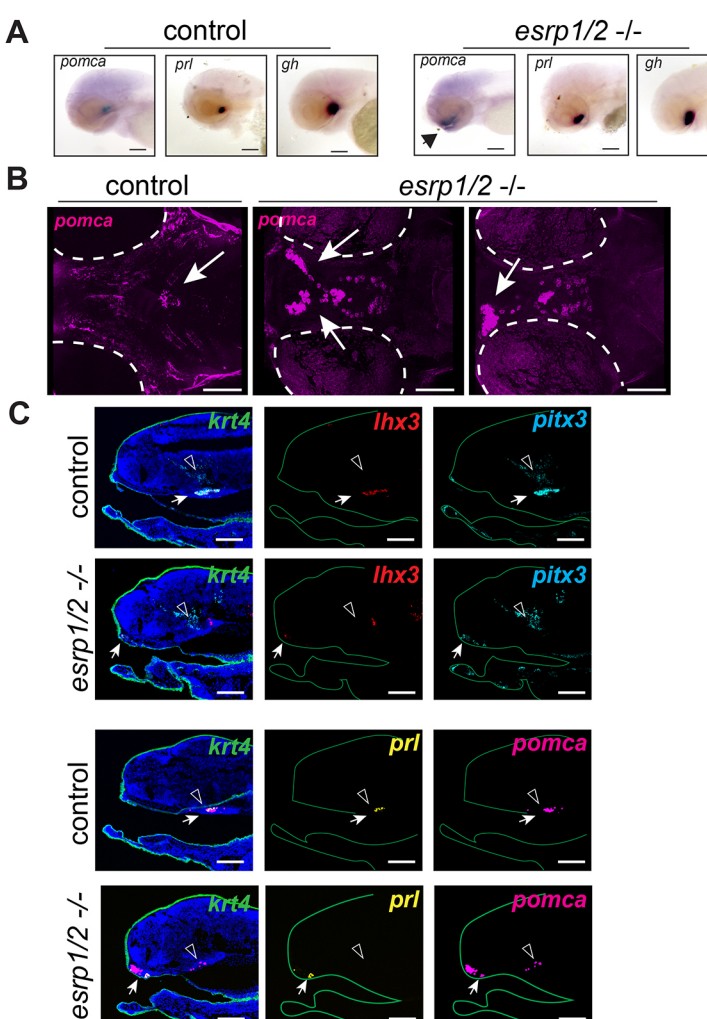

**Fig. 7. esrp1⁻/⁻,esrp2⁻/⁻ mutant zebrafish display adenohypophysis dysmorphogenesis.** (A) Whole-mount *in situ* hybridization of 3 dpf clutch-mate control and *esrp1⁻/⁻,esrp2⁻/⁻* mutant zebrafish. *pomca*, *prolactin* (*prl*) and *growth hormone* (*gh*) expression are displaced ventrally and rostrally in the *esrp1⁻/⁻, esrp2⁻/⁻* mutants relative to controls. Further, *pomca* expression is detected in the oral epithelium (arrow) of *esrp1⁻/⁻,esrp2⁻/⁻* mutants. *n*=5. (B) Dorsal view of 3D reconstruction of confocal imaging of *pomca* whole-mount RNAscope *in situ* hybridization of clutch-mate control and *esrp1⁻/⁻,esrp2⁻/⁻* mutant zebrafish at 3 dpf. While expression of *pomca* (magenta) in the control animals was observed as expected, *pomca* expression in *esrp1⁻/⁻,esrp2⁻/⁻* mutants was detected in a more ventral position and extended into the oral epithelium (arrows). *n*=2. (C) RNAscope *in situ* hybridization of midline sagittal sections of clutch-mate control and *esrp1⁻/⁻,esrp2⁻/⁻* mutant 3 dpf zebrafish. In controls, adenohypophysis (ADH, white arrows) markers *lhx3* (red), *pitx3* (cyan), *prl* (yellow) and *pomca* (magenta) are expressed as expected caudal to the ethmoid plate and adjacent to the hypothalamus (open arrowheads). In *esrp1⁻/⁻, esrp2⁻/⁻* mutants, *lhx3*, *pitx3*, *prl* and *pomca* are expressed within the oral cavity, highly rostral to normal positioning. Further, *pomca* expression in *esrp1⁻/⁻,esrp2⁻/⁻* mutants is normal within the hypothalamus but also more widespread than the other ADH genes. *krt4* expression was also detected (green) to identify oral epithelium. *n*=2 or 3. Scale bars: 100 μm.

Rathke's pouch of *Esrp1⁻/⁻;Esrp2⁻/⁻* mouse embryos is hypoplastic and improperly formed. Invagination of the surface/oral ectoderm is initiated and supported by Bmp4 and Fgf8/10 signals emanating from the adjacent infundibulum (Takuma et al., 1998). *Esrp1/2* are expressed in the surface/oral ectoderm and are known regulators of Fgf receptors, where the loss of *Esrp1/2* results in decreased expression of the IIIb receptor isoform of *Fgfr1*, *Fgfr2* and *Fgfr3*. This is also true in the invaginating prospective Rathke's pouch, as *Esrp1⁻/⁻; Esrp2⁻/⁻* embryos had decreased expression of the Fgfr1IIIb isoform. The functional significance of IIIb versus IIIc isoform expression remains unknown, though these isoforms contain different domains that likely change ligand/receptor kinetics. That reduced expression of Fgfr1IIIb in *Esrp1⁻/⁻; Esrp2⁻/⁻* embryos may lead to improper Rathke's pouch formation is supported by the previous finding that mice with specific ablation of the Fgfr2IIIb isoform have abnormal surface/oral ectoderm invagination and a failure to form a definitive pouch (De Moerlooze et al., 2000). In addition to changes in *Fgfr* isoform expression, Shh signaling and the Wnt pathway play an important role in Rathke's pouch formation (Potok et al., 2008; Rizzoti and Lovell-Badge, 2005; Treier et al., 1998). These pathways were previously shown to be reduced by *Esrp1/2* ablation (Lee et al., 2020); however, a direct or indirect role of Esrp1/2 on these pathways has yet to be identified.

Importantly, this improper morphogenesis of Rathke's pouch in *Esrp1⁻/⁻; Esrp2⁻/⁻* embryos is independent of pituitary cell specification, as early pituitary-dependent transcription factors are normally expressed. Interestingly, these pituitary progenitors do not go on to produce hormone-expressing cells, as these differentiated cells are absent in the mature anterior pituitary. The formation of the anterior lobe occurs through an epithelial-to-mesenchymal (EMT)-like transition of epithelial cells of the lumen of Rathke's cleft where the cells differentiate, delaminate and migrate, and contribute the parenchyma of the anterior lobe (Cheung et al., 2017). Esrp1/2 exert an epithelial-specific splicing program that is reverted upon initiation of EMT (Warzecha et al., 2010; Yang et al., 2016), and Esrp1/2 are implicated in developmental processes and organogenesis, as well as cancer progression (Derham and Kalsotra, 2023). It is possible that an improper splicing program in lumen epithelial cells of the *Esrp1/2* null embryos impairs the EMT-like transition necessary for the development of the anterior lobe. Here, we describe increased expression of Cdh1 in the ventral aspect of Rathke's pouch of *Esrp1⁻/⁻; Esrp2⁻/⁻* mice, which is consistent with suppressed EMT (Perez Millan et al., 2016, 2024). It is alternatively possible that the loss of an anterior lobe in the *Esrp1/2* null embryos is due to dysregulated proliferation or apoptosis of the migratory cells. In Fgfr2IIIb null mice, upregulated apoptosis was found in the newly formed anterior parenchyma, which may have led to the lack of an anterior lobe in these mice (De Moerlooze et al., 2000).

Aberrant EMT drives tumor aggressiveness in many cancers, and the key role of Esrp in regulating cancer invasion and metastasis is becoming increasingly appreciated (Wang et al., 2025). This is the

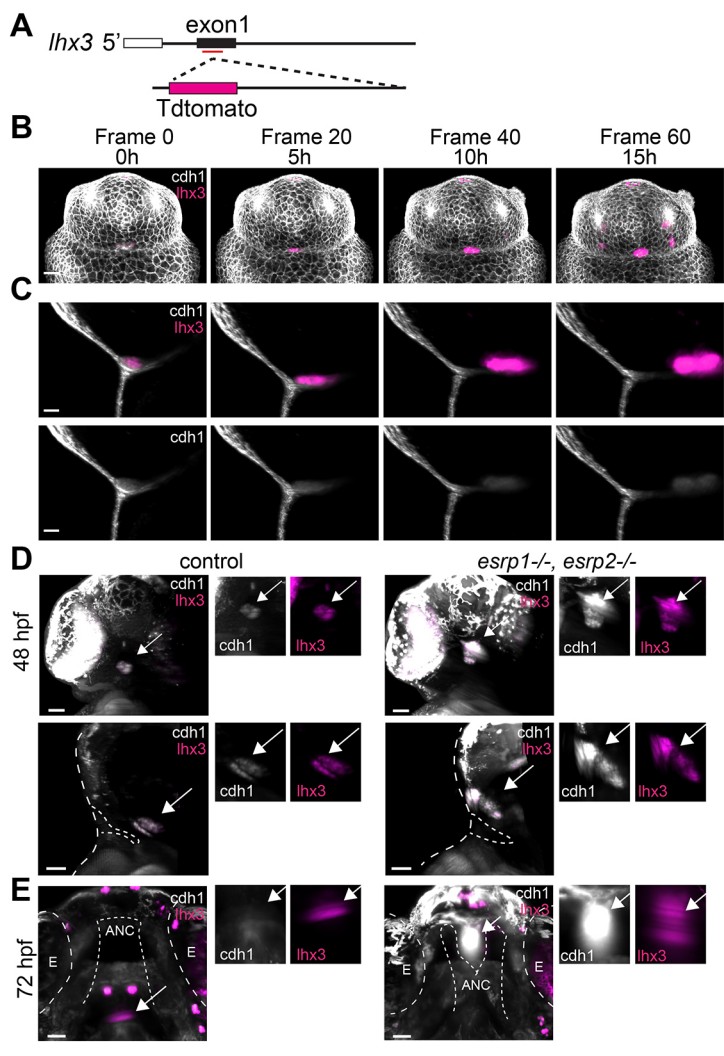

**Fig. 8. An *lhx3*:tdTomato reporter line shows impaired adenohypophysis anlage translocation in *esrp1⁻/⁻*,*esrp2⁻/⁻* mutant zebrafish.** (A) Schematic illustrating the generation of a zebrafish pituitary reporter line. (B-E) Light-sheet microscopy (LSM) imaging of zebrafish expressing *lhx3*:tdTomato (magenta) and *cdh1*:gfp (gray). (B) Frontal snapshots of LSM *in vivo* imaging from 27 to 42 hpf. (C) Snapshots of sagittal optical sections of LSM *in vivo* imaging from B. (D) Frontal and sagittal optical sections of LSM-generated images of clutch-mate control and *esrp1⁻/⁻, esrp2⁻/⁻* zebrafish at 48 hpf. Broken line delineates embryo surface and oral opening. (E) LSM-generated optical section through ventral view of *esrp1⁻/⁻, esrp2⁻/⁻* zebrafish at 72 hpf. Broken line delineates the eyes (E) and the anterior neurocranium cartilage (ANC). Arrows indicate the adenohypophysis anlage. *n*=3. Scale bars: 50 µm.

case for pituitary neuroendocrine tumors (pitNETs or pituitary adenomas) (Jia et al., 2015). In somatotroph adenomas, Esrp1 expression was correlated with E-cadherin expression, a negative correlator of EMT, tumor size and tumor invasiveness (Lekva et al., 2012). Further, a study of heterogeneity across pitNET subtypes found a worse clinical outcome associated with lower Esrp1

expression and impaired Esrp1-regulated splicing events (Huang et al., 2025). These findings, along with the data presented here, indicate a key role for Esrp in the generation and maintained stability of this important ectoderm-derived organ.

The embryonic development of the zebrafish pituitary (or adenohyohysis) is thought to be significantly different from that

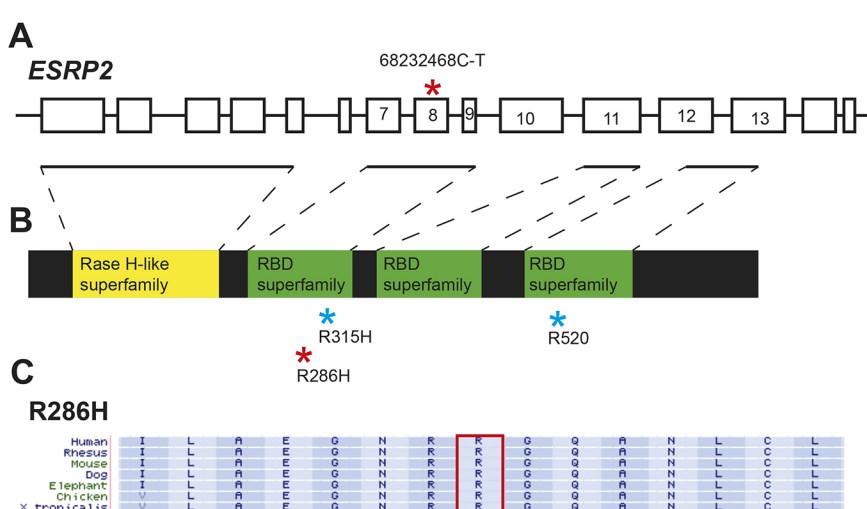

**Fig. 9. A deleterious *ESRP2* variant in an individual with hypopituitarism and cleft palate.** (A) A schematic of the human *ESRP2* gene. The red asterisk represents the location of the single nucleotide variant (SNV) identified in an individual with hypopituitarism and cleft palate. (B) Schematic of Esrp2 protein domains. The red asterisk represents the location of the resulting amino acid change within the RNA binding domain (RBD) of Esrp2. The blue asterisks represent the locations of previously identified *ESRP2* variants associated with cleft palate (Cox et al., 2018). (C) *ESRP2* amino acid sequences compared across species. The outlined region identifies the conserved arginine residue that is affected by the SNV.

of amniotes in that zebrafish pituitary morphogenesis begins with the internalization of the superficial adenohypophyseal placode and there is no obvious equivalent to Rathke's pouch. Additionally, cells of the internalizing anlage already express the hormones *prolactin* and *pomc,* suggesting terminal differentiation of lactotropes and corticotropes before organogenesis is completed (Herzog et al., 2003; Pogoda and Hammerschmidt, 2007). However, it has also been postulated that the invagination of the zebrafish stomodeum, which occurs in concert with the repositioning of the pituitary anlage represents a morphogenic equivalent between teleost and mammalian Rathke's pouch (Fabian et al., 2020; Herzog et al., 2003). This hypothesis has yet to be tested. Here, we demonstrate a shared requirement for *Esrp1/2* during pituitary morphogenesis in the mouse and zebrafish. This finding supports a conserved cellular mechanism for pituitary morphogenesis across these species.

A number of genes that are causative of congenital hypopituitarism (CH) have been identified; however, these account for fewer than 25% of cases (Camper et al., 2023; Fang et al., 2016). The previously unreported identification of *Esrp1/2* as a regulator of pituitary morphogenesis identifies it as a new gene candidate for the genetic diagnosis of CH. Genomic analysis using the known causative genes of CH failed to assign a genetic diagnosis for the individual described in this study. Our identification of a deleterious SNV in *ESRP2* in this individual suggests that *ESRP2* may be a new CH gene candidate. Functional testing of this specific variant in animal models, as well as additional analysis with larger CH cohorts, are needed to assign a causal link between CH and *ESRP2*. Further, targets of Esrp1/2 RNA regulation and splicing are promising candidates for being causative of CH. The consideration of alternatively spliced variants, and the utilization of transcriptome data for CH genetic diagnoses, has recently been highlighted (Camper et al., 2023). Our discovery of the requirement of *Esrp1/2* for pituitary morphogenesis across vertebrates underscores the developmental and clinical importance of alternative splicing.

## MATERIALS AND METHODS
### Animals
All animal experiments were performed in accordance with protocols approved by Children's Hospital of Philadelphia Institutional Animal Care and Use Committee. C57Bl/6J (WT) mice (*Mus musculus*) were obtained from the Jackson Laboratory. *Esrp1*$^{+/-}$; *Esrp2*$^{-/-}$ mice were received from Dr Russ Carstens (University of Pennsylvania, Philadelphia, PA, USA). Embryonic day 0.5 was considered to be noon on the day of the copulatory plug.

Zebrafish (*Danio rerio*) adults and embryos were maintained in accordance with approved institutional protocols at Children's Hospital of Philadelphia. Embryos were raised at 28.5°C in E3 medium (5.0 mM NaCl, 0.17 mM KCl, 0.33 mM CaCl$_2$ and 0.33 mM MgSO$_4$) with 0.0001% Methylene Blue. Embryos were staged according to standardized developmental time points by hours or days post-fertilization (hpf or dpf, respectively). All zebrafish lines used for experimentation were generated from the Tübingen strain. The *esrp1* and *esrp2* CRISPR mutants were generated as previously described (Carroll et al., 2020). The *lhx3*:tdTomato reporter line was generated using a previously described protocol (Auer et al., 2014; Hawkins et al., 2021). Briefly, a CRISPR guide targeting exon one near the ATG of *lhx3* was designed using Integrated DNA Technologies' Custom Alt-R CRISPR Cas9 guide RNA design tool. The donor plasmid containing the tdTomato sequence, an HSP70 promoter, and a linearization cut site was gifted by Dr Matt Harris. The guide, donor plasmid, plasmid linearization guide, lhx3-targeting guide and Cas9 protein were injected into one-cell stage zebrafish embryos. Proper insertion of the tdTomato donor DNA was determined by screening for heat shock-induced tdTomato expression followed by Sanger sequencing.

### Whole-mount *in situ* hybridization
Embryos were isolated at various time points and fixed in 4% formaldehyde at 4°C for 12-16 h. Subsequently, embryos were washed and stored in methanol. Whole-mount *in situ* hybridization and DIG-labeled riboprobes were synthesized as described previously (Thisse and Thisse, 2008). Whole-mount *in situ* hybridization colorimetric signal detection was performed using an alkaline phosphatase-conjugated anti-DIG antibody (Roche) and BM Purple AP substrate (Roche).

### RNAscope and Basescope *in situ* hybridization
Zebrafish and mouse embryos were fixed in 4% formaldehyde, taken through a sucrose gradient and cryo-embedded and sectioned. Probes were designed and purchased from ACD Bio, and hybridization and staining were performed according to the manufacturer's protocol. For whole-mount RNAscope *in situ* hybridization, zebrafish embryos were processed as described previously (Gross-Thebing et al., 2014) and imaged in 0.2% low-melt agarose. Whole-mounts and stained sections were imaged using a confocal microscope, where a *z*-stack was obtained and analyzed on ImageJ for *z*-stack maximum intensity projections. For Basescope, probes were designed and purchased from ACD Bio. Serial sagittal cryosections were taken through the mouse embryo heads and sections, where Rathke's pouch was most invaginated, were chosen for Basescope *in situ* hybridization. Hybridization and staining were performed according to the manufacturer's protocol, with 50% Gill's Hematoxylin I used as a counterstain. To quantify expression, the entire invaginated epithelium on a single section was manually identified, and the total number of green dots and red dots was counted under the microscope. As the invaginated area of *Esrp1/2*$^{-/-}$ mice were smaller than littermate controls, the total number of cell nuclei was counted and used to normalize the transcript number. Stained sections were imaged with a Leica bright-field microscope.

### Immunofluorescence
Immunofluorescence was performed on cryosections of mouse embryo heads using the following antibodies: anti-CD324 (E-Cad, Invitrogen 14-3249-82, lot 2892908, 1:100), anti-Lhx3 (Invitrogen PA5-117410, lot YE393309B, 1:250), anti-Sox2 (Abcam ab92494, lot 1093052-8, 1:100), anti-GH (Invitrogen PA5-79303, lot WB31189378, 1:200) and anti-POMC (Abcam ab210605, lot 102057-1, 1:500). All secondary antibodies were Alexa Fluor brand (Invitrogen). DAPI was used to stain nuclei. Images were taken using a confocal microscope (Leica SP8) and are presented as *z*-stack maximum projections using ImageJ.

### Light-sheet microscopy
Light-sheet fluorescent microscopy was performed on live zebrafish embryos using a Bruker Multi-View Selective-Plane Illumination Microscope (MuVi-SPIM). Embryos were anesthetized in 160 mg/l tricaine/MS-222 in E3 medium and mounted in 1.5% low-melt agarose. Embryos were maintained in E3 medium+tricaine during imaging. Images and movies were generated using Imaris software (Oxford Instruments).

### Clinical genomic data analysis
To examine individuals with pathogenic or likely pathogenic ESRP1/ESRP2 mutations in CHOP patients, we examined de-identified clinical records on a set of ~6900 patients affected with a range of clinical conditions with paired clinical exome sequencing results from the Division of Genomic Diagnostics (DGD). This analysis was performed in the institutional computing platform called Arcus, where clinical records and genome/exome sequencing data are available on subjects. We examined patients with Cleft Palate (CP, icd10 code - Q35) and Hypopituitarism (HPP, icd10 code - E23) annotations. The functional analysis of the variants and patient's whole-exome sequencing (in VCF format) was performed using ANNOVAR (Wang et al., 2010). The gnomAD version 4.1 (Karczewski et al., 2020) and dbNSFP database (Liu et al., 2020) were used while performing the variant analysis.

### Statistics
Statistical significance was determined by a two-tailed unpaired Student's *t*-test and data are presented as mean±s.d.

## Acknowledgements

We thank Christoph Seiler, Adele Donohue and the Aquatics Facility team at the Children's Hospital of Philadelphia for their excellent management of our zebrafish colonies and facilities. We also thank the CHOP Arcus team for provisioning computing infrastructure and virtual servers for data analysis, the Arcus Omics Science team for assistance in compiling exome sequencing files and clinical notes, and the IDDRC Biostatistics and Data Science core (HD105354) for technical consultation.

## Competing interests

The authors declare no competing or financial interests.

## Author contributions

Conceptualization: S.H.C.; Data curation: S.H.C., K.W.; Formal analysis: S.H.C.; Funding acquisition: E.C.L.; Investigation: S.H.C., S.S., A.S.R., P.W., M.U.A., L.T., K.W.; Methodology: K.W.; Project administration: E.C.L.; Writing – original draft: S.H.C., K.W.; Writing – review & editing: S.H.C., K.W., E.C.L.

## Funding

This work was supported by grants from the National Institutes of Health (R01DE032332 and R01DE027983 to E.C.L.) and by the Shriners Hospital for Children and Children's Hospital of Philadelphia Presidential Scholar Endowed Chair to E.C.L. Open Access funding provided by the National Institutes of Health. Deposited in PMC for immediate release.

## Data and resource availability

All relevant data and details of resources can be found within the article and its supplementary information.

## Peer review history

The peer review history is available online at https://journals.biologists.com/dev/lookup/doi/10.1242/dev.204636.reviewer-comments.pdf

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
