## [Peer Review File · Development (Cambridge, England)]

Genetic requirement for *Esrp1* and *Esrp2* in vertebrate pituitary morphogenesis

Shannon H. Carroll, Sogand Schafer, Ariella S. Richman, Peng Wang, Mian Umair Ahsan, Lisa Tsay, Kai Wang and Eric C. Liao
DOI: 10.1242/dev.204636

Editor: James M. Wells

Review timeline

Original submission:	9 January 2025
Editorial decision:	21 March 2025
First revision received:	3 September 2025
Accepted:	18 September 2025

Original submission

First decision letter

MS ID#: dev.204636

MS TITLE: Genetic requirement for *Esrp1/2* in vertebrate pituitary morphogenesis

AUTHORS: Shannon H. Carroll, Sogand Schafer, Ariella S. Richman, Peng Wang, Mian Umair Ahsan, Lisa Tsay, Kai Wang and Eric C. Liao

Dear Dr Liao,

I have now received all the referees' reports on the above manuscript, and have reached a decision. The referees' comments are appended below, or you can access them online: please go to:

As you will see, the referees express considerable interest in your work, but have some significant criticisms and recommend a substantial revision of your manuscript before we can consider publication. If you are able to revise the manuscript along the lines suggested, which may involve further experiments, I will be happy receive a revised version of the manuscript. Your revised paper will be re-reviewed by one or more of the original referees, and acceptance of your manuscript will depend on your addressing satisfactorily the reviewers' major concerns. Please also note that Development will normally permit only one round of major revision. If it would be helpful, you are welcome to contact us to discuss your revision in greater detail. Please send us a point-by-point response indicating your plans for addressing the referees' comments, and we will look over this and provide further guidance.

Please attend to all of the reviewers' comments and ensure that you clearly highlight all changes made in the revised manuscript. Please avoid using 'Tracked changes' in Word files as these are lost in PDF conversion. I should be grateful if you would also provide a point-by-point response detailing how you have dealt with the points raised by the reviewers in the 'Response to Reviewers' box. If you do not agree with any of their criticisms or suggestions please explain clearly why this is so.

Reviewer 1

Advance summary and potential significance to field

This straightforward study by Carroll et al. addresses the poorly understood clinical association between congenital hypopituitarism and orofacial clefting. The authors report that loss of function mutations in the splicing regulators *Esrp1/2* - previously shown to cause clefting in human, mouse, and zebrafish - also critically impair pituitary development in all three species. Using mouse models, they first show that both genes are expressed in the pertinent epithelia as Rathke's pouch invaginates and differentiates into the anterior/intermediate pituitary. Double *Esrp1/2* mutant mice lack the anterior pituitary; at earlier stages they show abnormal invagination of Rathke's pouch and an altered ratio of *Fgfr1* isoforms in the epithelium. A parallel investigation in zebrafish reveals similar expression patterns and morphogenetic defects in the developing pituitary, demonstrating conservation of function despite interspecies differences in how the pituitary forms. Finally, by mining existing exome data from their hospital, the authors identify a patient presenting with hypopituitarism and a cleft who bears a likely-pathogenic heterozygous missense variant in *ESRP2*. The analysis overall is logical, persuasive, and does not overreach. The work represents a solid advance that opens future directions for the research community and will have value for clinicians seeking mechanistic diagnoses for their patients.

Comments for the author

Issues to fix

Fig. 1: The low-magnification left panels in Fig. 1A and D appear to show nuclei alone, while those in 1B and C include the RNAscope channels. Assuming the RNAscope signals are not so low in A and D that they cannot be detected, it would be better to show the three-channel images for all four stages. Also note in the legend which nuclear stain was used.

Figs. 2-3: A schematic or additional anatomical notations would be helpful to orient the reader to these histological sections. Also note in the text and on the figure itself that the two different examples of the double mutant shown in 2A-B are different individuals.

Fig. 4: This finding of reduced *Fgfr1IIIb:IIIc* isoform ratio/content in the mutants roots the phenotype in an already-known function for *ESRP1/2* and appears to have been quantified as best as possible. However, the counterstain (not clear which it is) is very dark and somewhat obscures the BaseScope signal, and cell boundaries are difficult to make out. Adding a sentence to the methods to explain how these challenges were overcome would add confidence to the quantification.

Please also provide rationale for choosing to evaluate *Fgfr1* isoform expression when Moerlooze et al. (2000) reported a role for *Fgfr2* in development of Rathke's pouch.

Fig. 5: *esrp1* is clearly co-expressed with *pitx3* in this midline sagittal section, but the section itself is not readily recognizable; again, a schematic or additional anatomic markers would be helpful for orientation.

Fig. 5: The text notes that expression of both *esrp1* and *esrp2* was evaluated, but no data for *esrp2* are presented.

Fig. 6: In panel C, please mark the adenohypophysis vs. hypothalamic expression domains with letters or distinct arrow types

Fig. 7: At least one of the images in D/E should be split out to show the *lhx3:TdTomato* signal alone.

Panel letters for Fig. 3 should be added to the text.

Line 152: Add a semicolon after "closed"

Line 275: Change to "plus the *ESRP1* (in italics) gene"

Line 281: Change to "variants in *PERM1*"

Line 282: Change to "None of these three variants can explain..."

Additional suggestion: A quick PubMed search shows that *Esrp1/2* have been linked to progression of pituitary adenomas - this may be worth a mention in the discussion. How does this role align with the developmental role ascribed here?

Reviewer 2

Advance summary and potential significance to field

This is an interesting study that reports the effects of biallelic loss of function mutations in the related epithelial RNA splicing factors, ESRP1 and ESRP2, on pituitary gland development. Previous studies have shown that these factors have overlapping functions in craniofacial development and that loss of function causes orofacial clefting and cleft lip and palate. These features have been investigated by this group and others using mouse and zebrafish models. In humans, ESRP1 mutations have been reported in individuals with hearing impairment and ESRP2 mutations have been associated with cleft lip and palate. The role of these genes in pituitary gland development had not been investigated. The authors document expression of the *Esrp* genes in mouse and zebrafish pituitary development, illustrate severe defects in expansion of the pituitary primordium (mice), defects in migration of pituitary cells (fish), and suggest a mechanism involving FGF signaling. They also identify a novel ESRP2 patient with congenital hypopituitarism, expanding the phenotype associated with mutations in this gene. Overall, the work is well presented, the writing is fine, and the figures are clear and convincing. It represents a significant advance for the field. The paper could be improved by having a more thorough characterization of the mouse mutant and/or functional studies on the human variant.

Comments for the author

Developmental and cell specific expression in mice

Fig. 1 A, Is the *Esrp1/2* expression in Rathke's pouch only at e10.25, or does it extend into the nearby oral ectoderm? i.e. is it excluded from the *Shh* positive cells? At e11.5 and e12.5 it looks like it is throughout the oral ectoderm. Perhaps this could be clarified in the text of the results. Fig. 1 D, Are the *Esrp1* and 2 transcripts confined to the stem cell niche, or is it also detected in the parenchyma of the anterior lobe?

Mutant mouse analysis

The authors used pre-existing *Esrp1* and *Esrp2* knockout mice to assess the effect on pituitary development. Homozygous *Esrp1* mutants die at birth of cleft lip and palate. Homozygous *Esrp2* mutants are viable and grow normally. Homozygous *Esrp1*, *Esrp2* double mutants have severe pituitary hypoplasia; the anterior lobe appears absent. The same phenotype was found in *Esrp1* single mutants. In addition, the posterior lobe appears smaller than normal. This is a bit perplexing because *Esrp1* and 2 are not expressed there. The double mutants only express a trace amount of the hormone precursor POMC, and no GH at e17.5, a time when most hormone - producing cell types are readily detected.

They also report that the abnormality in pituitary development manifests early during Rathke's pouch development (e12.5). Apparently, a smaller amount of oral ectoderm tissue is earmarked for invagination to form Rathke's pouch. There appears to be some expression of the key pituitary transcription factors - *Lhx3* and *Prop1*, but *Prop1* is very weak and almost undetectable. A full assessment of critical pituitary transcription factors was not done. The authors suggest that the mechanism may involve failure to express the *Fgfr2IIIb* isoform robustly because FGF8 and FGF10 are critical for expansion of Rathke's pouch. It is likely that the mechanism is more complicated than simply affecting FGF signaling.

Fig. 2A. One of the *Esrp1/2* double mutants appears to have incomplete closure of the basisphenoid bone, but the other does not. Please indicate the penetrance of this feature, if known.

Also, it appears that the stem cell niche is dysmorphic. Since the *Esrp* genes regulate EMT in cancer cells, it would be useful to know whether there is evidence that pituitary stem cells have failed to undergo EMT. This may be obvious in higher magnification images, or by using some other antibodies for IHC.

Fig. 3B *Lhx3* transcripts are present, and *Prop1* transcripts are barely detectable. It appears that one of the mutants has remaining invaginated tissue in the oral ectoderm. This suggests that

the original patterning of Rathke's pouch by SHH signaling may have not been correct. There is likely to be some Shh expression because Lhx3 transcripts are present, but the pattern may be incorrect. Both the Lhx and Pitx genes are important for establishing pituitary fate. It is critical to examine expression of Pitx1 or Pitx2. Given the appearance of the pouch and oral ectoderm, I expect that Pitx2 is not expressed. Examining expression of these genes would help understand the underlying mechanism.

FGF, BMP, and SHH are all important for Rathke's pouch expansion. The authors have shown that FGF receptor splicing is altered. It should at least be mentioned that the *Esrp* deficiency could affect multiple signaling pathways because all of these signaling family receptors are alternately spliced, including the key pituitary transcription factor Pitx2. The *Fgfr11b* mutants have a more severe phenotype than the *Esrp* double mutants. This may be due to residual expression of the properly spliced isoform. The *Fgfr11b* mutant Rathke's pouch undergoes massive cell death, and the Pitx2 mutants have increased cell death. It may be valuable to conduct a TUNEL stain or activated caspase stain.

zebrafish analysis

The zebrafish studies are valuable because the mutants show failed pituitary cell migration. The authors developed a novel, tdTomato Lhx3 allele to trace the migration in controls and mutants.

Human variant

The authors screened clinical records of 6900 patients who had undergone clinical exome sequencing. There were 8 patients with hypopituitarism and an orofacial cleft. One had a missense variant, *ESRP2* p.R286H, in a critical domain, and no other candidate variants emerged. It would be helpful to clarify whether other patients with *ESRP2* variants are haploinsufficient (only one allele is mutated) or if they have biallelic mutations. It would also be valuable to test the function of the p.R286H variant.

First revision

Author response to reviewers' comments

Dear Editorial Team and Reviewers,

We would like to express deep appreciation for the thorough review of this work. The manuscript has benefited greatly from the comments and insights of the reviewers. We have made revisions accounting for all the comments as recommended. These changes are outlined in the itemized responses below and tracked in the red-lined version of the submitted manuscript.

Thank you very much for working with us on the publication of this paper.

Shannon Carroll and Eric Liao

ITEMIZED RESPONSES:

Comments from the Reviewers:

Reviewer 1: SUMMARY OF THE ADVANCE MADE IN THIS PAPER AND ITS POTENTIAL SIGNIFICANCE TO THE FIELD

This straightforward study by Carroll et al. addresses the poorly understood clinical association between congenital hypopituitarism and orofacial clefting. The authors report that loss of function mutations in the splicing regulators *Esrp1/2* - previously shown to cause clefting in human, mouse, and zebrafish - also critically impair pituitary development in all three species. Using mouse models, they first show that both genes are expressed in the pertinent epithelia as Rathke's pouch invaginates and differentiates into the anterior/intermediate pituitary. Double *Esrp1/2* mutant mice lack the anterior pituitary; at

earlier stages they show abnormal invagination of Rathke's pouch and an altered ratio of Fgfr1 isoforms in the epithelium. A parallel investigation in zebrafish reveals similar expression patterns and morphogenetic defects in the developing pituitary, demonstrating conservation of function despite interspecies differences in how the pituitary forms. Finally, by mining existing exome data from their hospital, the authors identify a patient presenting with hypopituitarism and a cleft who bears a likely-pathogenic heterozygous missense variant in ESRP2. The analysis overall is logical, persuasive, and does not overreach. The work represents a solid advance that opens future directions for the research community and will have value for clinicians seeking mechanistic diagnoses for their patients.

SUGGESTIONS TO AUTHORS

Issues to fix

Fig. 1: The low-magnification left panels in Fig. 1A and D appear to show nuclei alone, while those in 1B and C include the RNAscope channels. Assuming the RNAscope signals are not so low in A and D that they cannot be detected, it would be better to show the three-channel images for all four stages. Also note in the legend which nuclear stain was used.

Response: Thank you for bringing this to our attention. In 1A the left panel does include all channels, but the red and gray are difficult to distinguish at 10 and 20x. The left image in 1D has been replaced to show all channels, but again, the red and gray are challenging to see. The legend has been edited to indicate DAPI nuclear stain (Line 738).

Figs. 2-3: A schematic or additional anatomical notations would be helpful to orient the reader to these histological sections. Also note in the text and on the figure itself that the two different examples of the double mutant shown in 2A-B are different individuals.

Response: We have added additional annotations to Figs. 2 and 3 to orient the reader. We have also added annotation to denote the two different individuals. These annotations have been updated in the figure legends (Lines 747 and 763).

Fig. 4: This finding of reduced Fgfr1IIIb:IIIc isoform ratio/content in the mutants roots the phenotype in an already-known function for ESRP1/2 and appears to have been quantified as best as possible. However, the counterstain (not clear which it is) is very dark and somewhat obscures the BaseScope signal, and cell boundaries are difficult to make out. Adding a sentence to the methods to explain how these challenges were overcome would add confidence to the quantification.

Response: Thank you for understanding the challenges of this assay. We have added additional detail to the Methods on the counterstaining (hematoxylin) and how the quantification was performed (Lines 499-506).

Please also provide rationale for choosing to evaluate Fgfr1 isoform expression when Moerlooze et al. (2000) reported a role for Fgfr2 in development of Rathke's pouch.

Response: Our rationale for choosing to evaluate Fgfr1 isoforms is that FGFR1 variants are causative of Kallmann syndrome, which is associated with both pituitary malfunction and orofacial clefts. FGFR1 is also expressed during the development of Rathke's pouch (McCabe et al., 2011). However, we would predict the Fgfr2 isoform expression would also be disrupted in Rathke's pouch of the ESRP1/2 mutants, as both FGFR1 and FGFR2 have been shown to be ESRP1/2 targets.

Fig. 5: *esrp1* is clearly co-expressed with *pitx3* in this midline sagittal section, but the section itself is not readily recognizable; again, a schematic or additional anatomic markers would be helpful for orientation.

Response: Thank you for the excellent point. We have added a schematic of zebrafish anatomy to Figure 5 (now Fig. 6).

Fig. 5: The text notes that expression of both *esrp1* and *esrp2* was evaluated, but no data for *esrp2* are presented.

Response: Thank you. The text has been corrected (Line 241)

Fig. 6: In panel C, please mark the adenohypophysis vs. hypothalamic expression domains with letters or distinct arrow types

Response: A distinct arrowhead has been added to mark the hypothalamic expression domains and the figure legend has been updated (*Now Fig. 7*, Lines 827, 829).

Fig. 7: At least one of the images in D/E should be split out to show the *lhx3:TdTomato* signal alone.

Response: We have edited the figure to include images of each reporter signal alone. *Cdh1* is also expressed in the developing adenohypophysis, as has been previously reported (Fabian et al., 2020). (*Now Fig 8*)

Panel letters for Fig. 3 should be added to the text.

Response: Panel letters for Fig. 3 have been added to the Results section (Lines 161-169).

Line 152: Add a semicolon after "closed"

Response: The text has been corrected (Line 168).

Line 275: Change to "plus the *ESRP1* (in italics) gene"

Response: The text has been corrected (Line 336).

Line 281: Change to "variants in *PERM1*"

Response:

The text has been corrected (Line 342).

Line 282: Change to "None of these three variants can explain..."

Response: The text has been corrected (Line 343).

Additional suggestion: A quick PubMed search shows that *Esrp1/2* have been linked to progression of pituitary adenomas - this may be worth a mention in the discussion. How does this role align with the developmental role ascribed here?

Response: Thank you for the suggestion. We have added comments on the link between *Esrp1* and pituitary adenomas to the discussion section (Lines 414-423).

Reviewer 2: SUMMARY OF THE ADVANCE MADE IN THIS PAPER AND ITS POTENTIAL SIGNIFICANCE TO THE FIELD

This is an interesting study that reports the effects of biallelic loss of function mutations in the related epithelial RNA splicing factors, *ESRP1* and *ESRP2*, on pituitary gland development. Previous studies have shown that these factors have overlapping functions in craniofacial development and that loss of function causes orofacial clefting and cleft lip and palate. These features have been investigated by this group and others using mouse and zebrafish models. In humans, *ESRP1* mutations have been reported in individuals with hearing impairment and *ESRP2* mutations have been associated with cleft lip and palate. The role of these genes in pituitary gland development had not been investigated. The authors document expression of the *Esrp* genes in mouse and zebrafish pituitary development, illustrate severe defects in expansion of the pituitary primordium (mice), defects in migration of pituitary cells (fish), and suggest a mechanism involving FGF signaling. They also identify a novel *ESRP2* patient with

congenital hypopituitarism, expanding the phenotype associated with mutations in this gene. Overall, the work is well presented, the writing is fine, and the figures are clear and convincing. It represents a significant advance for the field. The paper could be improved by having a more thorough characterization of the mouse mutant and/or functional studies on the human variant.

SUGGESTIONS TO AUTHORS

Developmental and cell specific expression in mice

Fig. 1 A, Is the *Esrp1/2* expression in Rathke's pouch only at e10.25, or does it extend into the nearby oral ectoderm? i.e. is it excluded from the *Shh* positive cells? At e11.5 and e12.5 it looks like it is throughout the oral ectoderm. Perhaps this could be clarified in the text of the results.

Response: At e10.5 the *Esrp1/2* mRNA expression does extend to the oral ectoderm, as well as the entire surface ectoderm. This expression is difficult to see at low magnification. We have performed in situ hybridization for *Shh* expression and *Esrp1/2* mRNA are co-expressed in this region. We have added more information on the expression pattern to the text (Lines 122-123).

Fig. 1 D, Are the *Esrp1* and 2 transcripts confined to the stem cell niche, or is it also detected in the parenchyma of the anterior lobe?

Response: The *Esrp1/2* transcripts are also found in the parenchyma but to a lower degree than in the marginal zone. *Esrp1* expression is particularly low in the parenchyma. This detail has been added to the results section (Line 128).

Mutant mouse analysis

The authors used pre-existing *Esrp1* and *Esrp2* knockout mice to assess the effect on pituitary development. Homozygous *Esrp1* mutants die at birth of cleft lip and palate. Homozygous *Esrp2* mutants are viable and grow normally. Homozygous *Esrp1*, *Esrp2* double mutants have severe pituitary hypoplasia; the anterior lobe appears absent. The same phenotype was found in *Esrp1* single mutants. In addition, the posterior lobe appears smaller than normal. This is a bit perplexing because *Esrp1* and 2 are not expressed there. The double mutants only express a trace amount of the hormone precursor POMC, and no GH at e17.5, a time when most hormone-producing cell types are readily detected.

They also report that the abnormality in pituitary development manifests early during Rathke's pouch development (e12.5). Apparently, a smaller amount of oral ectoderm tissue is earmarked for invagination to form Rathke's pouch. There appears to be some expression of the key pituitary transcription factors - *Lhx3* and *Prop1*, but *Prop1* is very weak and almost undetectable. A full assessment of critical pituitary transcription factors was not done. The authors suggest that the mechanism may involve failure to express the *Fgfr2IIIb* isoform robustly because FGF8 and FGF10 are critical for expansion of Rathke's pouch. It is likely that the mechanism is more complicated than simply affecting FGF signaling.

Response: We agree that the *Esrp1/2* mutant phenotype begins with less tissue contributing Rathke's pouch. We also agree with the reviewer that a difference in the expression pattern of key morphogens, such as *Shh*, could explain the impaired oral ectoderm invagination and Rathke's pouch formation in the *Esrp* KOs. We have performed cursory IF staining for *Shh*, *Ptch1*, and *Fgf8* but did not observe any differences to warrant further investigation. We speculate that as *Esrp1/2* are regulators of alternative splicing, their absence may be causing a change in the expression pattern of specific isoforms, such that cannot be detected by antibody staining. It may also be that differences in morphogen expression patterns are not apparent at the developmental timepoints we evaluated, or would be better determined by whole mount staining, rather than histological sections.

We acknowledge with the reviewer that although Fgf receptors are established targets of *Esrp1/2*, there are other targets that are likely contributing to the KO phenotype, and we hope to identify these with further investigation. We have added additional discussion regarding these possible mechanisms to the discussion (Lines 388-392).

Fig. 2A. One of the *Esrp1/2* double mutants appears to have incomplete closure of the basisphenoid bone, but the other does not. Please indicate the penetrance of this feature, if known.

Response: Thank you for your close observation. Although we saw this incomplete closure in some of our histological sections, it was variable, and we did not analyze a large enough number of embryos to make a determination as to the penetrance.

Also, it appears that the stem cell niche is dysmorphic. Since the *Esrp* genes regulate EMT in cancer cells, it would be useful to know whether there is evidence that pituitary stem cells have failed to undergo EMT. This may be obvious in higher magnification images, or by using some other antibodies for IHC.

Response: We have performed some additional IF and RNAscope in situ hybridization. We found higher expression of eCadherin (*Cdh1*) in our *Esrp1/2* double KO in the inferior region of Rathke's pouch, which is consistent with impaired EMT. We have presented these findings in the Results (Lines 204-226) and in a new figure (Figure 5). RNAscope with probes for *Zeb1*, *Zeb2* resulted in low expression in Rathke's pouch at E12.5, with no difference between control and *Esrp1/2* KO littermates.

Fig. 3B *Lhx3* transcripts are present, and *Prop1* transcripts are barely detectable. It appears that one of the mutants has remaining invaginated tissue in the oral ectoderm. This suggests that the original patterning of Rathke's pouch by SHH signaling may have not been correct. There is likely to be some *Shh* expression because *Lhx3* transcripts are present, but the pattern may be incorrect. Both the *Lhx* and *Pitx* genes are important for establishing pituitary fate. It is critical to examine expression of *Pitx1* or *Pitx2*. Given the appearance of the pouch and oral ectoderm, I expect that *Pitx2* is not expressed. Examining expression of these genes would help understand the underlying mechanism.

Response: To expand our assessment of critical pituitary transcription factors, we have performed additional in situ hybridization experiments for *Isl1* and *Pitx2* (Fig. 3A,B). We did not find an observable difference in the expression of these transcription factors, other than the overall difference in morphology in the *Esrp1/2* null embryos. Additionally, we have changed the color of *Prop1* expression from red to yellow so that its pattern can be more easily visualized (Fig. 3C). Taken together, these data support the hypothesis that *Esrp1/2* regulate the morphogenesis of Rathke's pouch, rather than pituitary cell determination.

FGF, BMP, and SHH are all important for Rathke's pouch expansion. The authors have shown that FGF receptor splicing is altered. It should at least be mentioned that the *Esrp* deficiency could affect multiple signaling pathways because all of these signaling family receptors are alternately spliced, including the key pituitary transcription factor *Pitx2*.

The *Fgfr3* mutants have a more severe phenotype than the *Esrp* double mutants. This may be due to residual expression of the properly spliced isoform. The *Fgfr3* mutant Rathke's pouch undergoes massive cell death, and the *Pitx2* mutants have increased cell death. It may be valuable to conduct a TUNEL stain or activated caspase stain.

Response: Cursory IF staining of activated caspase at E11 showed no significant difference between WT and *Esrp1/2* KO littermates; however, it is possible that differences could be present at a different developmental timepoint. We have added comments regarding the possible role of additional signaling pathways in the *Esrp1/2* KO phenotype to the Discussion (Lines 388-392). *Pitx2* alternative splicing is a very interesting hypothesis, however whether *Pitx2* is a direct target of *Esrp1/2* remains unknown.

zebrafish analysis

The zebrafish studies are valuable because the mutants show failed pituitary cell migration. The authors developed a novel, tdTomato *Lhx3* allele to trace the migration in controls and mutants.

Human variant

The authors screened clinical records of 6900 patients who had undergone clinical exome sequencing. There were 8 patients with hypopituitarism and an orofacial cleft. One had a missense variant, ESRP2 p.R286H, in a critical domain, and no other candidate variants emerged. It would be helpful to clarify whether other patients with ESRP2 variants are haploinsufficient (only one allele is mutated) or if they have biallelic mutations. It would also be valuable to test the function of the p.R286H variant.

Response: Cases where ESRP2 are associated with orofacial cleft or congenital deafness, the pathogenic variant is biallelic.

Second decision letter

MS ID#: dev.204636R1

MS TITLE: Genetic requirement for *Esrp1/2* in vertebrate pituitary morphogenesis

AUTHORS: Shannon H. Carroll, Sogand Schafer, Ariella S. Richman, Peng Wang, Mian Umair Ahsan, Lisa Tsay, Kai Wang and Eric C. Liao

Dear Dr Liao,

I am happy to tell you that your manuscript has been accepted for publication in Development, pending our standard publication integrity checks.

Reviewer 1*Advance summary and potential significance to field*

This straightforward study by Carroll et al. addresses the poorly understood clinical association between congenital hypopituitarism and orofacial clefting. The authors report that loss of function mutations in the splicing regulators *Esrp1/2* - previously shown to cause clefting in human, mouse, and zebrafish - also critically impair pituitary development in all three species. Using mouse models, they first show that both genes are expressed in the pertinent epithelia as Rathke's pouch invaginates and differentiates into the anterior/intermediate pituitary. Double *Esrp1/2* mutant mice lack the anterior pituitary; at earlier stages they show abnormal invagination of Rathke's pouch and an altered ratio of *Fgfr1* isoforms in the epithelium. A parallel investigation in zebrafish reveals similar expression patterns and morphogenetic defects in the developing pituitary, demonstrating conservation of function despite interspecies differences in how the pituitary forms. Finally, by mining existing exome data from their hospital, the authors identify a patient presenting with hypopituitarism and a cleft who bears a likely-pathogenic heterozygous missense variant in ESRP2. The analysis overall is logical, persuasive, and does not overreach. The work represents a solid advance that opens future directions for the research community and will have value for clinicians seeking mechanistic diagnoses for their patients.

Comments for the author

In Figure 8D, *cdh1*:GFP appears much more intensely expressed in the *lhx3*⁺ cells of the mutant compared with the control. Is this consistent across samples or representative of truly elevated endogenous *cdh1* expression? If so, it may be worth commenting on how this aligns with their other finding of higher *Cdh1* in Rathke's pouch of the mouse mutant.

Figure 9B: Correct "Rase H-like" to "RNase H-like"

All of my previous comments have been adequately addressed.

Reviewer 2

Advance summary and potential significance to field

This is an interesting study that reports the effects of biallelic loss of function mutations in the related epithelial RNA splicing factors, ESRP1 and ESRP2, on pituitary gland development. Previous studies have shown that these factors have overlapping functions in craniofacial development and that loss of function causes orofacial clefting and cleft lip and palate. These features have been investigated by this group and others using mouse and zebrafish models. In humans, ESRP1 mutations have been reported in individuals with hearing impairment, and ESRP2 mutations have been associated with cleft lip and palate. The role of these genes in pituitary gland development had not been investigated. The authors document expression of the *Esrp* genes in mouse and zebrafish pituitary development, illustrate severe defects in expansion of the pituitary primordium (mice), defects in migration of pituitary cells (fish), and suggest a mechanism involving FGF signaling. They also identify a novel ESRP2 patient with congenital hypopituitarism, expanding the phenotype associated with mutations in this gene. Overall, the work is well presented, the writing is fine, and the figures are clear and convincing. It represents a significant advance for the field. The revised manuscript has been improved by having a more thorough characterization of the mouse mutant.

Comments for the author

None. The authors have done substantially more work that helps understand the mouse phenotype.